# Generating dense packings of hard spheres by soft interaction design

T. Maimbourg[1*], M. Sellitto[2,1], G. Semerjian[3], F. Zamponi[3]

**1** The Abdus Salam International Centre for Theoretical Physics, Strada Costiera 11, 34151 Trieste, Italy
**2** Dipartimento di Ingegneria, Università degli Studi della Campania "Luigi Vanvitelli", Via Roma 29, I-81031 Aversa, Italy
**3** Laboratoire de Physique Théorique de l'Ecole Normale Supérieure, PSL University, CNRS, Sorbonne Universités, 24 rue Lhomond, 75231 Paris Cedex 05, France
* tmaimbou@ictp.it

August 6, 2018

## Abstract

**Packing spheres efficiently in large dimension $d$ is a particularly difficult optimization problem. In this paper we add an isotropic interaction potential to the pure hard-core repulsion, and show that one can tune it in order to maximize a lower bound on the packing density. Our results suggest that exponentially many (in the number of particles) distinct disordered sphere packings can be efficiently constructed by this method, up to a packing fraction close to $7\,d\,2^{-d}$. The latter is determined by solving the inverse problem of maximizing the dynamical glass transition over the space of the interaction potentials. Our method crucially exploits a recent exact formulation of the thermodynamics and the dynamics of simple liquids in infinite dimension.**

# 1 Introduction

The sphere packing problem consists in finding the densest arrangements of equal-sized spheres in the $d$-dimensional Euclidean space. This is a simply defined geometrical problem with intriguing and sometimes unexpected connections to other areas of mathematics, natural sciences and engineering. In this paper we shall study the sphere packing problem in the limit of large space dimension $d$. Far from being an abstract problem, this asymptotic limit is known, since Shannon's pioneering work, to be connected to the practical problem of designing error correcting codes in communication technology [1]. Another motivation for this study comes from statistical physics. In this context the large $d$ limit provides a theoretical framework in which computations are simplified, and one can study analytically the complex phase diagram of interacting colloids [2–4].

In the sphere packing problem, the degrees of freedom are the positions of the centers of a set of impenetrable spheres. It can be thus naturally viewed as a Constraint Satisfaction Problem (CSP) [5]: the positions of the spheres form the variables, which have to fulfill a constraint for each pair of particles, namely that the distance between their centers is larger than their diameter. The largest packing density separates, in this setting, the satisfiable (SAT) regime in which all the constraints can be simultaneously satisfied (a packing exists) from the unsatisfiable (UNSAT) regime in which they cannot (no packing exists).

The term CSP is usually encountered in the context of problems with discrete variables, for instance graph $q$-coloring or $k$-satisfiability. These two examples are NP-complete decision problems [6] (for $q, k \geqslant 3$); to characterize their typical complexity an important research effort has been devoted to the study of random ensembles of such CSPs, for instance the coloring of Erdös-Rényi random graphs, or the satisfiability of random formulas [7, 8]. These problems can be recast as mean-field spin-glasses models [9, 10], the randomness in their construction washing out any Euclidean geometry. The techniques developed in the statistical mechanics of disordered systems community, in particular the replica and cavity method, thus apply to these random CSPs as well, and have brought a lot of important quantitative and qualitative results [11–14]. In particular, the location of the SAT-UNSAT phase transition can be computed in this statistical mechanics framework. Even more importantly, it is found that other phase transitions, in the SAT phase, affect the structure of the subset of solutions in the configuration space. In particular, starting from an underconstrained situation and increasing the density of constraints, one generically encounters a "dynamic" phase transition where the solution space breaks apart into a very large number of disjoint clusters. At this dynamic transition, uniform sampling of solutions becomes exponentially hard in the number of variables [15].

These same techniques can also be used to study *disordered* hard sphere packings. At variance with spin-glasses and random CSP, sphere packings have no quenched disorder, but a phenomenon of self-induced disorder upon increasing density allows one to treat them in a similar way [3, 16–18]. Another difference is the finite-dimensionality $d$ of the packing problem, to be contrasted with the mean-field character of random CSPs. However in the recent years a controlled $d \to \infty$ theory has been set up [3, 4], and the limiting theory was shown to share several qualitative properties with random CSPs. This analogy has been very fruitful and has allowed to treat in a unified way these seemingly different phenomena.

Consider a system of $N$ hard spheres in a $d$-dimensional volume $V$ with periodic boundary conditions, in the thermodynamic limit where $N, V \to \infty$ at constant packing fraction $\varphi =$

$N v_d/V$ ($v_d$ being the volume of a unit-diameter sphere), and let us recall the most important results obtained from this approach, in the large $d$ limit (taken after the thermodynamic limit), that are relevant for present discussion:

1. For packing fractions $\varphi < \varphi_d$, with $\varphi_d = 4.8067... \, d \, 2^{-d}$, the liquid can be equilibrated in a time that scales polynomially in the number of particles $N$ [19]. Equivalently, the (exponentially many in $N$) available packings at $\varphi < \varphi_d$ can be sampled uniformly in polynomial time in $N$. On the contrary, for $\varphi > \varphi_d$, the sampling takes a time growing exponentially in $N$.

2. The equilibrium configurations produced at $\varphi = \varphi_d$ can still be compressed out of equilibrium, up to a density $\varphi_{j,d}$ where their (out of equilibrium) pressure diverges and further compression becomes impossible (also called the *jamming transition*). The precise value of $\varphi_{j,d}$ has not been computed, but it can be approximately located around $\varphi_{j,d} \approx 7.4 \, d \, 2^{-d}$ [20].

3. Despite being exponentially hard to sample, disordered packings exist up to a density scaling as $\varphi_{\text{GCP}} = (\ln d) \, d \, 2^{-d}$, that corresponds to an equilibrium SAT-UNSAT phase transition (the acronym GCP standing for Glass Close Packing) [3].

It should be noted that, by construction, this approach is focused on disordered packings (defined as those that can be constructed by adiabatic compression from the liquid state), and therefore it does not provide any information on lattice packings [3].

Let us now describe the main idea developed in the present paper. The phase transitions described above explicitly referred to the Gibbs-Boltzmann distribution which is *uniform* over the solutions of the CSPs, i.e. the configurations that simultaneously satisfy all the constraints. In the particle systems language, this means a purely hard-sphere interaction, all positions of the spheres that lead to an overlap between them are assigned an infinite energy and thus forbidden, but all other configurations have the same vanishing energy. Consider now a situation where the constraints of the CSP are still imposed in a strict way, but the probability measure on the solutions can be biased to favour some solutions and disfavour others (for particles, this means a pairwise potential energy infinite below the diameter of the particles, but arbitrary at larger distance). By definition, the location of the (equilibrium) SAT-UNSAT transition is not altered by this modification, but the other structural phase transitions in the satisfiable phase will in general be. In particular, one can hope to increase in this way the density $\varphi_d$ of the dynamic transition, hence to enlarge the regime of parameters where solutions can be obtained efficiently (even if no more uniformly); recent works in the context of discrete CSP can be interpreted along this line [21–23]. Here we shall follow this idea in the context of particle systems, and look for the soft part of the interparticle potential that, in the mean-field infinite-dimensional analytical formulation, yields the highest possible packing fraction of the dynamic transition.

The outline of the rest of the paper is as follows. In Sec. 2 we provide a short review of existing results (rigorous and not) on the sphere packing problem. In Sec. 3, we briefly recall the theoretical background and the main formulas relating the dynamical liquid-glass transition to the interaction potential of simple liquids. Next, we describe the numerical strategy we used for solving the "inverse problem" of finding the potential that maximizes $\varphi_d$, and its actual implementation for the special case of piecewise constant potentials. In Sec. 4 we present the main results of our work, and discuss their relevance. In Sec. 5, we close with some remarks and a discussion about improvements and other possible applications.

# 2 Review of previous results on the sphere packing problem

For completeness, we present here a brief review of some results, originating both from the mathematical and physical literature, concerning upper and lower bounds on the density of the densest sphere packings. More detailed reviews can be found, respectively, in [1, 24], and in [3, 4, 25].

## 2.1 Rigorous results

The sphere packing problem in physical dimensions $d = 2$ and $3$ is an emblematic example in mathematics of how a seemingly obvious fact may require very elaborated concepts to be actually proven. While the optimal packing fraction in $d = 3$ can be easily guessed, as Kepler did, by stacking as close as possible triangular layers[1], it took almost four centuries and a genuine *tour de force* to prove that the conjecture is indeed correct [27–29]. There are further reasons that make the sphere packing challenging. First, as soon as we move above $d = 3$ there appears a sensitive dependence of the optimal packings on the space dimensionality. Every dimension has its own geometric oddities and, unlike the most favorable case $d \leqslant 3$, what is known about the densest packings in a certain dimension $d$ is generally useless for other dimensions, even those very close to $d$. The recent proof for the densest packing in $d = 8$ [30] and its extension to $d = 24$ [31] are a lucky exception in this respect. Second, when $d$ grows large the unusual geometrical features of high dimensional spaces become more pronounced [32]: the volume of the unit diameter sphere vanishes exponentially with respect to the volume of the unit hypercube (see Figure 1 for a simple illustration). The empty spaces become predominant and optimal packings in large dimensions cannot be very dense: in this asymptotic limit only upper and lower bounds are known (see [24, 25] for a more detailed review). A "greedy" argument shows that the volume occupied by a saturated packing[2] must satisfy $\varphi \geqslant 2^{-d}$. This lower bound, obtained by Minkowski in 1905, was improved by a linear $d$ factor by Rogers in 1947 [33], and it has been difficult to do much better since then: Vance's 2011 bound is $\varphi \geqslant (6/e) d \, 2^{-d}$ (for $d$ divisible by 4) [34]. Only very recently Venkatesh has dramatically improved on it by a constant of 65963 instead of $6/e$, and by a $\ln(\ln d)$ factor for a nongeneric/sparse set of dimensions [35]. While most of these lower bounds are non-constructive, very recently Moustrou [36] proved that packings with density $\varphi \geqslant 0.89 \ln(\ln d) \, d \, 2^{-d}$ can be constructed with $\exp(1.5 \, d \ln d)$ binary operations.

One might also be interested in counting the number of possible packings, or better, their entropy. Lebowitz and Penrose in 1964 [37] have proven that the virial series expansion of the entropy is convergent for $\varphi < 0.14467... \, 2^{-d}$, providing in this regime the exact result for the excess entropy per particle (over the one of a Poisson process, or an ideal gas of non-interacting particles) in the Gibbs-Boltzmann uniform measure[3]:

$$s^{\text{ex}}(\varphi) = -\frac{2^d \varphi}{2} \ . \tag{1}$$

This result was extended to higher densities in [38], where it was shown that at packing fraction $\varphi = \ln(2/\sqrt{3}) \, d \, 2^{-d}$, the entropy density of the Gibbs measure is $s^{\text{ex}} \geqslant -2^d \varphi$. Note

---

[1]This triangular lattice itself was only proven in 1940 to be the densest in dimension $d = 2$ [1, 26]

[2]A packing is saturated when there is no room for adding an extra sphere.

[3]See [38], second equation on page 11, for the definition.

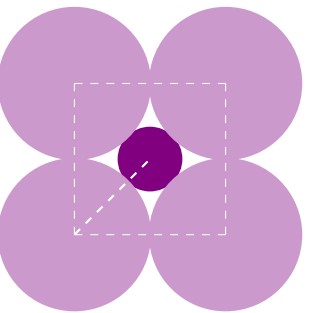

Figure 1: The four circle paradox [32]. A small circle of radius $\sqrt{2}-1$ can fit the space left by four circles of unit radius. One can repeat the same construction in the $d$-dimensional Euclidean space by using $2^d$ hyperspheres of unit radius that surround a hypersphere of radius $\sqrt{d}-1$. Above $d=4$ the darker sphere grows larger than the pale ones. In fact, the volume of the room left by the $2^d$ touching spheres tends asymptotically to $2^d$, as for large $d$ the volume of a hypersphere becomes vanishingly small, $V_d(r) = \pi^{d/2} r^d / \Gamma(d/2+1)$, and tends to concentrate on its boundary, $V_d(1-\epsilon)/V_d(1) = (1-\epsilon)^d \to 0$ for any small $\epsilon > 0$ as $d \to \infty$.

that this bound is compatible with the validity of Eq. (1), and the missing factor 2 is likely due to technical details of the proof (see section 2.2).

It is important to emphasize that the known best upper bound [39] on the densest packing fraction scales as $2^{-0.5990...d}$, and therefore its ratio with the best lower bound grows exponentially in $d$, thus leaving a huge uncertainty in the location of the optimal packing density for large $d$. Moreover, it is by no means obvious that for large $d$ there exist "universal features" of optimal packings, and that they should necessarily have a lattice structure [25]. In physical language, lattice packings represent crystalline configurations, disordered packings would correspond to liquid or glassy configurations. Disordered packings in large space dimension are therefore very interesting and their study could shed further light on this problem.

## 2.2 Physics results

We will approach the sphere packing problem from a statistical physics perspective. In this context, hard-sphere systems have been used since long to describe the behaviour of gases, most notably by van der Waals in 1873 [40], whose equation of state provided the first example of a thermodynamic phase transition, by Metropolis et al. in 1953 [41], who introduced the famous "Metropolis algorithm" to study the liquid equation of state in $d=2$, and by Alder and Wainwright in 1957 [42] who showed the existence of a first order phase transition between the liquid and the crystal states in $d=3$. Subsequently, starting from Bernal in 1960 [43], hard spheres have also been used to model the geometric structure of denser phases, such as liquids or crystals.

More recently, physicists have formulated a series of conjectures on the large $d$ behavior of the hard sphere liquid and glass phases, that according to physics standard are believed to be exact, although they have not yet been proven rigorously. Frisch et al. in 1985 [44] (see also [45–47]) have investigated the hard sphere fluid in the limit $d \to \infty$ and concluded that its entropy is formally given by the second-virial equation of state, Eq. (1), up to densities as high as $\varphi < (e/2)^{d/2} 2^{-d}$. However, a careful study of the liquid dynamics in $d \to \infty$, initiated in [2] and later fully developed in [19, 48], shows that the dynamics becomes arrested at a

critical packing fraction $\varphi_{\mathrm{d}} = 4.8067...\, d\, 2^{-d}$ [19]. Above this critical density, the liquid is a collection of metastable glassy phases, or in other words, the allowed hard sphere configurations are organised in disjoint *clusters* in phase space, precisely as it happens in satisfiability problems [14]. Based on this analogy and on the results of [15], it is believed that sampling uniformly configurations of $N$ hard spheres takes a time increasing polynomially in $N$ for $\varphi < \varphi_{\mathrm{d}}$, and exponentially in $N$ for $\varphi > \varphi_{\mathrm{d}}$. One can use advanced statistical mechanics techniques (mostly the replica method, see [3, 4] for a review of these results), to show that the glassy clusters of configurations keep existing up to a density scaling as $\varphi_{\mathrm{GCP}} = (\ln d) d\, 2^{-d}$, above which the liquid phase cannot be adiabatically continued to any other meaningful phase. Above this density, either no packings exist, or a first order transition to a non-translationally invariant phase (most likely a crystal) takes place. Note that numerical simulations in spatial dimensions 3 to 12 partially support these conjectures [49, 50].

For a related statistical mechanics approach to lattice packings in large $d$, see [51], and [25, 52] for a different statistical mechanics approach to disordered packings.

# 3 Setup of the problem

## 3.1 High-dimensional formalism for pairwise interacting particles

As discussed in the Introduction, the Gibbs-Boltzmann uniform measure over hard sphere configurations can be biased by adding an arbitrary interaction potential to the hard core. From a physical point of view, even a microscopically small bias in the hard-core potential is generally known to induce large-scale non-trivial phase behaviours, such as water-like anomalies and multiple liquid phases [53–56], reentrancy and multiple glass phases [57–62], and instabilities towards disordered or periodic microphases [63–65].

In this paper we will restrict ourselves to pairwise isotropic interactions between particles, with a potential energy for two particles at distance $r$ of the form[4]

$$v(r) = \begin{cases} \infty \,, & r < 1 \,, \\ v_+(r) \,, & r > 1 \,, \end{cases} \tag{2}$$

where $v_+(r)$ is an arbitrary finite function that converges to 0 faster than $r^{-d-1}$ when $r \to \infty$, in such a way that the second virial coefficient remains finite. In the limit $d \to \infty$, the interaction potential should be scaled as [66, 67]

$$v(r) = \begin{cases} \infty \,, & h < 0 \,, \\ \bar{v}(h) \,, & h > 0 \,, \end{cases} \,, \qquad h = d(r-1) \,, \tag{3}$$

where $h$ represents the typical $O(1/d)$ fluctuations of the liquid particles during their time evolution. Indeed, it turns out that the distance between neighbouring particles that are effectively interacting is the diameter of the spheres plus this fluctuation. Note that in statistical physics, the inverse temperature $\beta = 1/T$ multiplies the potential in the Gibbs measure; here, to simplify the notations, we will include this factor in the definition of the potential (hence,

---

[4]The hard sphere diameter is usually set to 1 in the packing problem, meaning that all lengths are measured relatively to it.

our $v(r)$ corresponds to $\beta v(r)$ in usual notations). Finally, it is convenient to introduce a rescaled packing fraction $\widehat{\varphi}$ according to

$$\varphi = \widehat{\varphi} \frac{d}{2^d} \ , \tag{4}$$

with $\widehat{\varphi}$ of order one in the dense liquid regime where the dynamical arrest happens [4, 19].

With these definitions, it has been shown in [3, 19, 67] that for an arbitrary extra potential $\bar{v}(h)$:

- The excess entropy of the liquid is given by

$$s^{\mathrm{ex}} = \frac{d\widehat{\varphi}}{2} \left[ -1 + \int_0^\infty \mathrm{d}h\, e^h [e^{-\bar{v}(h)} - 1] \right] \ . \tag{5}$$

- The dynamical transition density is defined by the condition

$$\frac{1}{\widehat{\varphi}_{\mathrm{d}}} = \max_\Delta \mathcal{F}[\Delta | \bar{v}(h)] \ , \tag{6}$$

with

$$\mathcal{F}[\Delta | \bar{v}(h)] = -\Delta \int_{-\infty}^\infty \mathrm{d}y\, e^y \ln[q(\Delta, y)] \frac{\partial q(\Delta, y)}{\partial \Delta} \ ,$$

$$q(\Delta, y) = \int_0^\infty \mathrm{d}h\, e^{-\bar{v}(h)} \frac{e^{-\frac{(h-y-\Delta/2)^2}{2\Delta}}}{\sqrt{2\pi\Delta}} \ . \tag{7}$$

For $\widehat{\varphi} < \widehat{\varphi}_{\mathrm{d}}$, configurations of the Gibbs-Boltzmann measure can be sampled in polynomial time in $N$, above it, exponential time in $N$ is needed.

- The value of $\Delta$ for which the maximum of $\mathcal{F}[\Delta | \bar{v}(h)]$ is attained corresponds to the long time limit of the mean square displacement in the dynamically arrested glass phase at $\widehat{\varphi}_{\mathrm{d}}$:

$$\Delta = \lim_{t \to \infty} \frac{d}{N} \left\langle \sum_{i=1}^N [\boldsymbol{r}_i(t) - \boldsymbol{r}_i(0)]^2 \right\rangle \ , \tag{8}$$

where $\boldsymbol{r}_i(t)$ is the position of particle $i$ at time $t$, and the dynamics start in equilibrium at time $t = 0$ [67].

- The disordered phase exists up to the density $\varphi_{\mathrm{GCP}} = (\ln d)d\, 2^{-d}$, independently of the biasing potential [3].

The results recalled in section 2.2 correspond to the pure hard-sphere case, $\bar{v}(h) = 0$. One can then naturally ask the question whether a judicious choice of the biasing potential $\bar{v}(h)$ could push the dynamical transition to higher values, thus facilitating the task of finding hard sphere configurations. This question has already been positively answered in the simplest case in which $\bar{v}(h)$ is a square-well short-ranged attractive potential [66, 68],

$$\bar{v}(h) = \begin{cases} -U_0 & \text{for } 0 \leqslant h \leqslant \sigma_0 \\ 0 & \text{for } h > \sigma_0 \end{cases} \ . \tag{9}$$

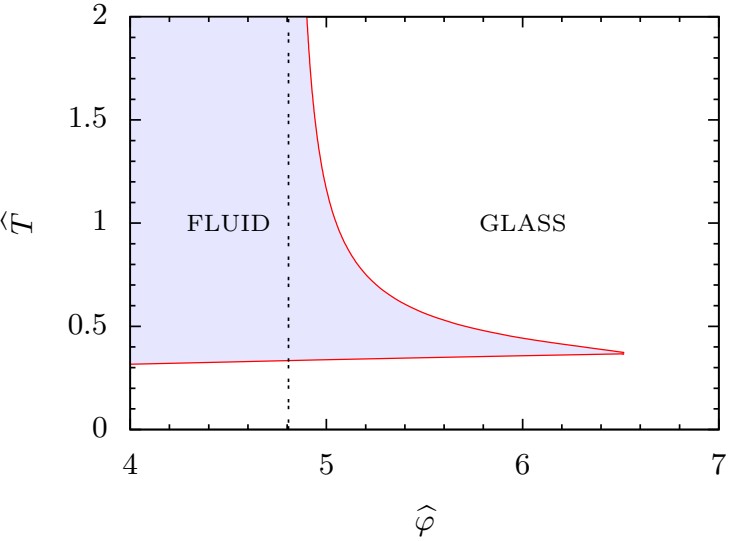

Figure 2: A section of the phase diagram of a hard-sphere system with an additional square-well attraction of strength $U_0$ for a value of the width of the well $\sigma_0 = 0.0293$. $\widehat{T} = 1/U_0$ is the temperature scaled by the depth of the potential energy well. The vertical dotted line at $\widehat{\varphi} = 2^d \varphi/d \simeq 4.8067$ represents the purely hard-sphere glass transition (in the absence of attraction).

Figure 2 shows the dynamical transition line obtained for this potential, for a value of the attraction range $\sigma_0 = 0.0293$ at which the effect of the potential is the most pronounced. One sees indeed that $\widehat{\varphi}_{\mathrm{d}}$ has a maximum, as a function of $U_0$, that is significantly larger than its purely hard-sphere value (corresponding to $U_0 = 0$), causing a reentrance phenomenon between the liquid and glass phases.

This effect generally suggests that the introduction of a suitable additional interaction leads the ergodic liquid region to penetrate quite deeply in the nonergodic glass phase, thereby increasing the density at which one can realize hard core configurations (disordered packings), although with a biased (non-uniform) measure. It is interesting to note that the dense packings constructed in this way are quite robust against noise or statistical fluctuations due to their thermodynamic stability and their finite entropy density, according to Eq. (5).

In the following, we show that it is possible to extend further the re-entrance of the liquid region deeper in the glass phase by adding a suitable potential to the hard-core repulsion. This is determined by solving the inverse problem of maximizing the dynamical liquid-glass transition $\widehat{\varphi}_{\mathrm{d}}$, given by Eq. (6), over the set of possible interaction potentials $\bar{v}(h)$. By doing so we find that the reentrancy tip can reach a packing fraction value near $7\, d\, 2^{-d}$. Interestingly, the liquid state with the highest density turns out to lie on a multicritical manifold along with several qualitatively different glass states, whose number is close to the number of lengthscales that enter in the definition of the interaction potential.

## 3.2 The numerical strategy

The implicit dependency of $\widehat{\varphi}_d$ on $\bar{v}(h)$ in Eq. (6) makes it difficult to approach the solution of the inverse problem by purely analytical means, so we turn to a numerical optimization algorithm. To handle this variational problem we consider a trial family of potentials, namely the piecewise constant ones with $n$ constant steps, each depending on a pair of variables, the width $\sigma_i$ and the height $-U_i$ of the step[5]. Explicitly, we generalize the single-step ($n = 1$) potential given in Eq. (9) to

$$
\bar{v}(h) = \begin{cases}
-U_0 & \text{for } 0 \leqslant h \leqslant \sigma_0 \\
-U_1 & \text{for } \sigma_0 < h \leqslant \sigma_0 + \sigma_1 \\
\dots & \\
-U_{n-1} & \text{for } \sum_{i=0}^{n-2} \sigma_i < h \leqslant \sum_{i=0}^{n-1} \sigma_i \\
0 & \text{for } h > \sum_{i=0}^{n-1} \sigma_i
\end{cases} \tag{10}
$$

Increasing $n$ enlarges the family of potentials on which the maximization of $\widehat{\varphi}_d$ is performed, hence yields better and better lower bounds on its optimal value, that should be reached in the formal limit $n \to \infty$ that allows to describe arbitrary continuous potentials (as the optimal one is expected to be).

There are several advantages in the choice of the variational ansatz presented in Eq. (10). As the widths of the steps are free parameters, unlike setting e.g. a fixed discretization grid, it accounts more easily for both slow and fast-varying regions of the potential: a slow-varying region (of small derivative) can be roughly described as a single step whereas fast-varying regions require a finer description including many steps. Furthermore, this ansatz is "range-agnostic" in the sense that one does not have to make any assumption about the range of the potential, as the ansatz should adapt itself when converging to the densest potential. Finally, the function $q(\Delta, y)$ in Eq. (7) is known *analytically* for this choice of $\bar{v}(h)$, in terms of error functions which are readily and efficiently implemented numerically:

$$
q(\Delta, y) = \frac{1}{2} \left\{ 1 + \text{erf}(\tau_{n-1}) + \sum_{i=0}^{n-1} e^{U_i} \left[ \text{erf}(\tau_{i-1}) - \text{erf}(\tau_i) \right] \right\} ,
$$

$$
\text{with} \quad \tau_i = \frac{\Delta/2 + y - \sum_{j=0}^{i} \sigma_j}{\sqrt{2\Delta}} ,
$$

(11)

while the expression of $\partial q / \partial \Delta$ has a similar form with instead a sum of Gaussian functions. Hence there is no need for an additional numerical integration procedure to compute the function $q(\Delta, y)$ for each choice of its parameters, which speeds up significantly the numerical study.

In the following we present our results for different values of $n$, i.e. the largest $\widehat{\varphi}_d$ obtained within this variational ansatz and the corresponding potential $\bar{v}(h)$. As said above, increasing the number of steps leads to monotonically increasing packing fractions; however, we shall see that within our numerical accuracy no sensible improvement can be actually achieved by going beyond $n = 4$ or 5. The extrapolated packing fraction saturates at a value near $7\,d\,2^{-d}$.

---

[5]Most steps will turn out to be negative, hence the choice of the minus sign.

Depending on the number of steps $n$, hence on the number of free parameters $2n$ (the width and depth of each step) we have followed two numerical strategies to get the highest $\widehat{\varphi}_\mathrm{d}$. First, one may perform numerically an extensive search by computing $\widehat{\varphi}_\mathrm{d}$ for all possible values of each of the $2n$ parameters of the potential, for a fine enough discretization of the parameter space. This has been done for both single-step and two-step ansätze. For more than 4 parameters the numerical computation turns out to be prohibitively long, and is therefore not suitable to investigate the problem in full generality. This strategy is thus mainly used as a benchmark for the next one at $n = 1$ and 2. Indeed, in a more systematic way, we can proceed with a gradient descent in the parameter space of the function $\widetilde{\mathcal{F}}[\bar{v}(h)] = \max_\Delta \mathcal{F}[\Delta|\bar{v}(h)]$, where the gradient is computed over the $2n$ parameters defining $\bar{v}(h)$. With a slight abuse of terminology we shall call in the following "landscape" the function $\widetilde{\mathcal{F}}$; the gradient descent procedure relies on the assumption that this landscape is smooth and not riddled with local minima. Note from Eq. (6) that minima of $\widetilde{\mathcal{F}}$ correspond to maxima of the packing fraction $\widehat{\varphi}_\mathrm{d}$.

## 3.3 Case of a single step

The case of a single step coincides with a previous study by two of us [66,68], where the phase diagram of the system around its glass transition was studied in detail. The landscape being two dimensional here, it is easy to visualize. Besides, one can perform a full sampling of the parameters' space due to the small number of parameters.

First, for $U_0 < 0$ (a single positive step), we have sampled numerically the landscape $\widehat{\varphi}_\mathrm{d}(U_0, \sigma_0)$ and it turns out to be monotonically decreasing (as a function of $\sigma_0$ for $U_0 < 0$ fixed) from the pure hard sphere case $\sigma_0 = 0$, for which $\widehat{\varphi}_\mathrm{d}^\mathrm{HS} \simeq 4.8067$. This potential is known in the colloids' literature as the square-shoulder potential [62]. The function $\mathcal{F}(\Delta|U_0, \sigma_0)$ has a single maximum over $\Delta$ [66,68], computing the gradient with respect to $(U_0, \sigma_0)$ at this point yields a simple descent that always converges towards $(U_0, \sigma_0) \to (0, 0)$ when the initial condition satisfies $U_0 < 0$. The physical interpretation is rather straightforward: the single positive step acts as a strictly repulsive region, thereby increasing the distances between all spheres, thus lowering the overall packing fraction.

For a negative step ($U_0 > 0$), the potential gets an attractive behaviour at short distances, leaving hope for an increase of the packing fraction. It is more usually called a square-well potential. A fine sampling of the parameters' space is plotted in figure 3. The evolution of the packing fraction is rather smooth, except at one very sharp line (yellow in figure 3) where the global maximum lies. Close to this region, the function $\mathcal{F}(\Delta|U_0, \sigma_0)$ develops a second maximum, while away from it, it has only a single maximum. This was already noted in [66,68], where the secondary maximum is interpreted as a metastable glass phase (named either attractive or repulsive depending on the respectively small or large value of the corresponding mean-square displacement within a cage, $\Delta$). When there are several maxima of $\mathcal{F}$, the dynamics selects the highest maximum, i.e. the dynamical transition occurs at the lowest possible density [69,70]. Interestingly, the very sharp line of high density is the *transition line of the two glass phases* (called hereafter glass-glass, or *GG line*), namely the line where both maxima have the same value, meaning that the fluid to glass transition is here a tricritical dynamical transition line separating the fluid phase and the two glass phases. An important consequence of the appearance of this GG line is that the gradient $\nabla \mathcal{F}$ in terms of the potential parameters is discontinuous on this line: the line separates locally the phase diagram in two sides where the function $\mathcal{F}$ has two maxima. On one side the attractive

maximum dominates while on the other it is the repulsive one that dominates. Therefore on each side the gradient is computed on the dominant maximum, resulting in a discontinuity on the GG line where the gradient must jump from one side's value to the other. The landscape is thus singular at this line.

The consequence on a gradient descent is then clear: a standard gradient descent of the parameters $(U_0, \sigma_0)$ will always reach some point on the line $(U_0^{(l)}, \sigma_0^{(l)})$, depending on its initial condition. Yet, it will not be able to find the global maximum since once on the line, the discontinuity of the gradient will make the trajectory point $(U_0, \sigma_0)$ jump off the GG line, resulting in a decrease of the packing fraction. This emphasizes the need of an improved gradient descent able to follow the GG line, performing a gradient descent within it. Another crucial aspect of this critical region is that the packing fraction varies significantly close to the line, so that this line may be easily missed by a rougher landscape sampling (this phenomenon also shows up in Figure 2 where the tip of the fluid phase is very narrow). This is why a full sampling becomes quickly impractical as the number of steps $n$ grows.

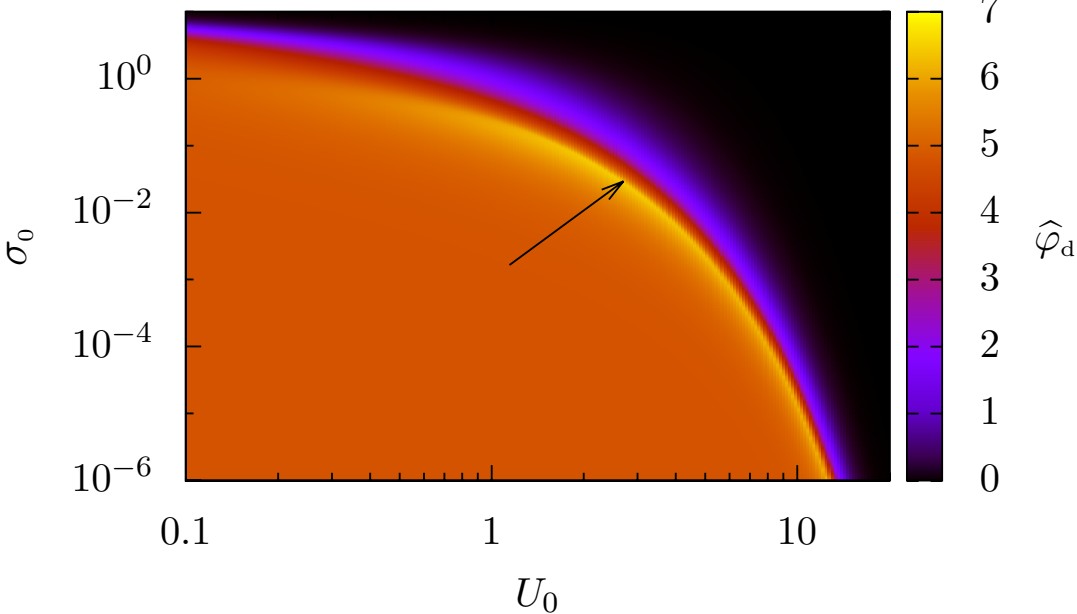

Figure 3: Plot of $\widehat{\varphi}_d(U_0, \sigma_0)$ for a the square-well potential $U_0 > 0$ ($n = 1$). The arrow points to the $n = 1$ maximal packing fraction.

In this $n = 1$ case, both the full sampling and the gradient descent agree on the global maximum, found within the GG line. The gradient descent has been performed in the improved way able to follow the GG line, described in the next section. The values of the best-packing potential are given in table 2. The plot of the potential is shown in figure 5.

## 3.4   The gradient descent

For higher values of $n$, we tackle the problem of finding the highest packing fraction $\widehat{\varphi}_d$ by performing a gradient descent on the function $\widetilde{\mathcal{F}}(\boldsymbol{M}) = \max_{\Delta} \mathcal{F}(\Delta | \boldsymbol{M})$ of the potential

parameters $\boldsymbol{M} = (U_0, \sigma_0, \ldots, U_{n-1}, \sigma_{n-1})$, to find its absolute minimum (if possible). The principle of the standard gradient descent is that at a point in parameter space $\boldsymbol{M}$, after having computed the location $\Delta^*(\boldsymbol{M})$ of the global maximum of $\mathcal{F}(\Delta|\boldsymbol{M})$, that we assume for now to be unique and which constitutes the value of $\Delta$ in the following, one follows the best direction to minimize $\widetilde{\mathcal{F}}$. Expanding

$$\widetilde{\mathcal{F}}(\boldsymbol{M} + \boldsymbol{\delta}) = \widetilde{\mathcal{F}}(\boldsymbol{M}) + \boldsymbol{\delta} \cdot \boldsymbol{\nabla}\widetilde{\mathcal{F}}(\boldsymbol{M}) + O(\|\boldsymbol{\delta}\|^2) \ , \tag{12}$$

the best direction is locally given by $\boldsymbol{\delta} \propto -\boldsymbol{\nabla}\widetilde{\mathcal{F}}(\boldsymbol{M})$, thus the gradient descent consists in updating $\boldsymbol{M} \longrightarrow \boldsymbol{M} - \varepsilon\boldsymbol{\nabla}\widetilde{\mathcal{F}}(\boldsymbol{M})$ which decreases the value of $\widetilde{\mathcal{F}}$ for small enough $\varepsilon$. Iterating this strategy one ends up in a local minimum depending on the initial starting point, if the function is smooth. As one can see in figure 3, the function $\widetilde{\mathcal{F}}$ varies rather on a logarithmic scale of the parameters, and it is faster to use a gradient descent in terms of the logarithm of the potential parameters, replacing $\boldsymbol{M}$ by $\ln \boldsymbol{M}$ (the logarithm being taken component-wise). As mentioned in the last section, there are several difficulties associated to the standard gradient descent:

- First, the function may have several local minima, and depending on the initial conditions of the descent, one may therefore not find the global minimum. Besides, performing the logarithmic gradient descent prevents any change of sign of the parameters during the evolution. This is dealt with by running gradient descents with many different initial conditions (including different signs of the $U_i$), see the algorithm in appendix A.

- Second, we empirically find that the descents always evolve towards manifolds with several maxima of $\mathcal{F}(\Delta)$ with equal value, as mentioned in the case of the $n = 1$ GG line (with only two maxima) in last section. On these GG manifolds, the gradient is discontinuous, rendering the gradient descent pointless. The way around is that when the gradient descent encounters such discontinuities, it must switch to a gradient descent *within this manifold* where the largest packing fractions are found, as described below.

The general idea is the following: locally the GG manifold can be defined as being orthogonal to some subspace. The components of the gradients in this subspace calculated in different maxima are different (which is at the origin of the discontinuity). However, because the gradient finds the best slope locally, its components within the manifold (locally an Euclidean subspace with as many dimensions as the number of maxima minus one, as shown in appendix B) correspond to the best slope within the manifold, which we are looking for. The optimal direction within the manifold as well as a summary of the algorithm are detailed in appendices A and B.

## 4 Numerical results for several steps

Numerical investigations were performed for $n \leqslant 6$ steps. Let us now comment on each case.

### 4.1 Two steps

For two steps, a fine sampling of the parameters' space can still be achieved in a reasonably short time. This method agrees with the best-packing potential found within the gradient descent. Fixing one of the two steps around some point, a plot similar to figure 3 can be

obtained, and shows again that there exist very sharp regions of maximal densities, that necessitate a fine sampling to be detected. In parallel, one can use the gradient descent method, first performing a loose sampling of the parameters' space from which 50 high-density initial conditions are drawn. Here, the gradient descent has 4 outcomes depending on the sign of the two steps $U_0$, $U_1$ at the initial condition. Note that this is due to the fact that the logarithmic gradient descent prevents changes of sign of the potential parameters, and there appears to be a single global maximum for each possibility. The data is displayed in table 1.

- $U_0 < 0$, $U_1 < 0$: it converges to the pure hard sphere potential $U_0 = U_1 = 0$, $\sigma_0 = \sigma_1 = 0$.

- $U_0 < 0$, $U_1 > 0$: it converges to the $n = 1$ best packer for the second step, while $\sigma_0 \to 0$.

- $U_0 > 0$, $U_1 < 0$: it converges to the $n = 2$ best packer with $\widehat{\varphi}_{\mathrm{d}} \simeq 6.729$ with 2 equal-height maxima of $\mathcal{F}$.

- $U_0 > 0$, $U_1 > 0$: it converges to the $n = 2$ secondary density maximum with 3 equal-height maxima of $\mathcal{F}$.

One sees from these results and the extensive sampling that the function $\mathcal{F}$ develops up to three local maxima. There exist regions with two maxima of $\mathcal{F}$ having the same value, and even three in another region. An important remark is that when these regions exist, the descent always converges towards a region with two maxima of same value, and then following the GG manifold, it either finds a maximal density and stops, or finds its way to the coexistence of three maxima of equal height, where it evolves within this new manifold and finds the secondary maximum of the packing fraction's landscape.

We note that the $n = 2$ best packer has a negative first step and a small positive second step. Physically, this means the potential is attractive at short distances while it has some repulsive behaviour at larger distances. This second step has the effect of deepening the well created by the first one, leading to a better "trapping" of the neighbouring spheres. There is a trade-off in this second step between the improved stickiness at short distance and the larger-ranged repulsive effect that must remain small.

## 4.2 More than two steps

For $n > 2$ the extensive search is computationally too heavy. With the experience gained from the $n = 1$ and 2 cases, we now rely on the gradient descents only.

The trial potential of Eq. (10), for a given $n$, contains as special cases (when some $\sigma_i$'s vanish, or when some $U_i = U_{i+1}$) all the potentials with a strictly smaller number of steps. As a consequence its landscape contain local minima arising from these limits, that have a higher value of $\widetilde{\mathcal{F}}$ (smaller value of the packing fraction) than the global optimum for this value of $n$. It turns out that, indeed, most of the gradient descent runs from the initial conditions get stuck to potentials that have one very small step for example, or small variations from a $n' < n$ potential minimum (or secondary minimum). Nevertheless, several gradient descents for each $n$ could escape from these and find strictly better potentials. The densest packings for each $n$ are displayed in table 2.

As seen in the case of a lower number of steps, the gradient descents tend to flow to manifolds with the highest possible number of maxima of $\mathcal{F}$ with the same value. An exception

| Initial condition | $U_0$ | $\sigma_0$ | $U_1$ | $\sigma_1$ | $\widehat{\varphi}_{\rm d}$ | Maxima |
|---|---|---|---|---|---|---|
| $U_0 > 0\,,\ U_1 < 0$ | 2.15 | 0.0536 | -0.0636 | 2.29 | 6.729 | 2 |
| $U_0 > 0\,,\ U_1 > 0$ | 5.24 | 0.00187 | 0.843 | 0.0719 | 6.608 | 3 |
| $U_0 < 0\,,\ U_1 > 0$ | 0 | 0 | 2.70 | 0.0292 | 6.515 | 2 |
| $U_0 < 0\,,\ U_1 < 0$ | 0 | 0 | 0 | 0 | 4.807 | 1 |

Table 1: Densities reached by the gradient descent depending on the starting point, at $n = 2$. The last column indicates the number of equal-height maxima of the function $\mathcal{F}$.

| $n$ | $U_0$ | $\sigma_0$ | $U_1$ | $\sigma_1$ | $U_2$ | $\sigma_2$ | $U_3$ |
|---|---|---|---|---|---|---|---|
| 1 | 2.70 | 0.0293 | 0 | 0 | 0 | 0 | 0 |
| 2 | 2.15 | 0.0536 | -0.0636 | 2.29 | 0 | 0 | 0 |
| 3 | 4.54 | 0.00369 | 0.485 | 0.160 | -0.0821 | 1.94 | 0 |
| 4 | 5.83 | 0.000971 | 1.15 | 0.0353 | 0.182 | 0.194 | -0.0840 |
| 5 | 5.93 | 0.000868 | 1.10 | 0.0440 | 0.129 | 0.225 | -0.0896 |
| 6 | 5.93 | 0.000876 | 1.30 | 0.0252 | 0.328 | 0.0989 | 0.0145 |

| $n$ | $\sigma_3$ | $U_4$ | $\sigma_4$ | $U_5$ | $\sigma_5$ | $\widehat{\varphi}_{\rm d}$ | Maxima |
|---|---|---|---|---|---|---|---|
| 1 | 0 | 0 | 0 | 0 | 0 | 6.516 | 2 |
| 2 | 0 | 0 | 0 | 0 | 0 | 6.729 | 2 |
| 3 | 0 | 0 | 0 | 0 | 0 | 6.924 | 4 |
| 4 | 1.805 | 0 | 0 | 0 | 0 | 6.952 | 5 |
| 5 | 1.46 | -0.0409 | 0.645 | 0 | 0 | 6.959 | 5 |
| 6 | 0.235 | -0.0920 | 1.42 | -0.0257 | 0.865 | 6.966 | 6 |

Table 2: Best-packing potentials found by the gradient descent procedure as a function of the number of steps $n$. The last column indicates the number of equal-height maxima of the function $\mathcal{F}(\Delta)$ for the corresponding potential.

is $n = 2$, where the densest packing is not found in the region with 3 glass phases. A plot of $\mathcal{F}$ for the potential yielding the densest packing found at $n = 4$ is displayed in figure 4.

The shape of the corresponding potentials is plotted in figure 5. Their qualitative features are quite similar: a sticky attraction at short distances that continuously turns into a repulsive behaviour at the largest length scales, arising from a positive "tail" of the potential. As $n$ increases, they seem to converge to a limit shape. In terms of the packing fraction $\widehat{\varphi}_{\rm d}$ at the dynamic transition, we see that already at these small values of $n$ the enhancement quickly converges; the gain with respect to the $n = 1$ best packer is quite limited and becomes of the order of $10^{-2}$ once $n > 3$. For these reasons, going to a higher number of steps is rather pointless. Besides, the computational complexity increases (for example, at $n = 6$ even the loose grid sampling becomes quite inefficient in order to find relevant starting points). Actually, this strategy of running several ($\approx 50 - 70$) gradient descents from an initial loose sampling of the landscape was only performed for $n \leqslant 5$. In the $n = 6$ case, because the potential has already well converged onto a limit shape, we simply ran 5 gradient descents from an initial potential given by the best-packing $n = 5$ potential where one step was splitted into two new steps, as it is likely that the best $n = 6$ potential should be close in shape to the optimal one found with $n = 5$.

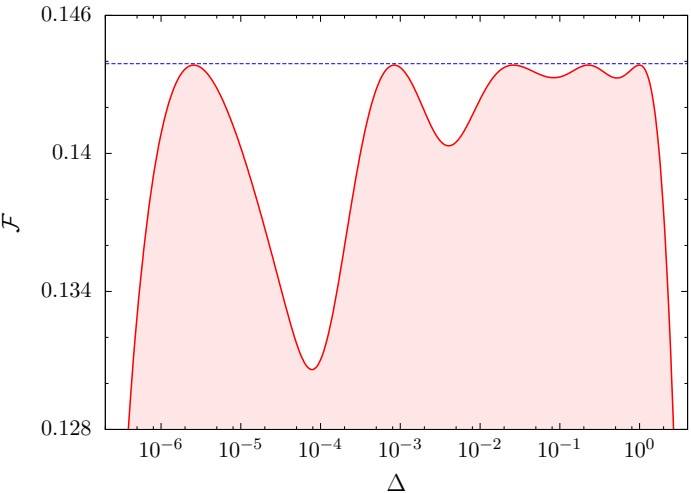

Figure 4: The function $\mathcal{F}[\Delta|\bar{v}(h)]$ at the liquid-glass transition for a system of hard spheres interacting with a four-step potential maximizing the packing fraction at the transition. There are 5 maxima with the same height, corresponding to 5 distinct glass phases. $\mathcal{F}[\Delta|\bar{v}(h)]$ goes monotonically to zero for both $\Delta \to 0$ and $\Delta \to \infty$.

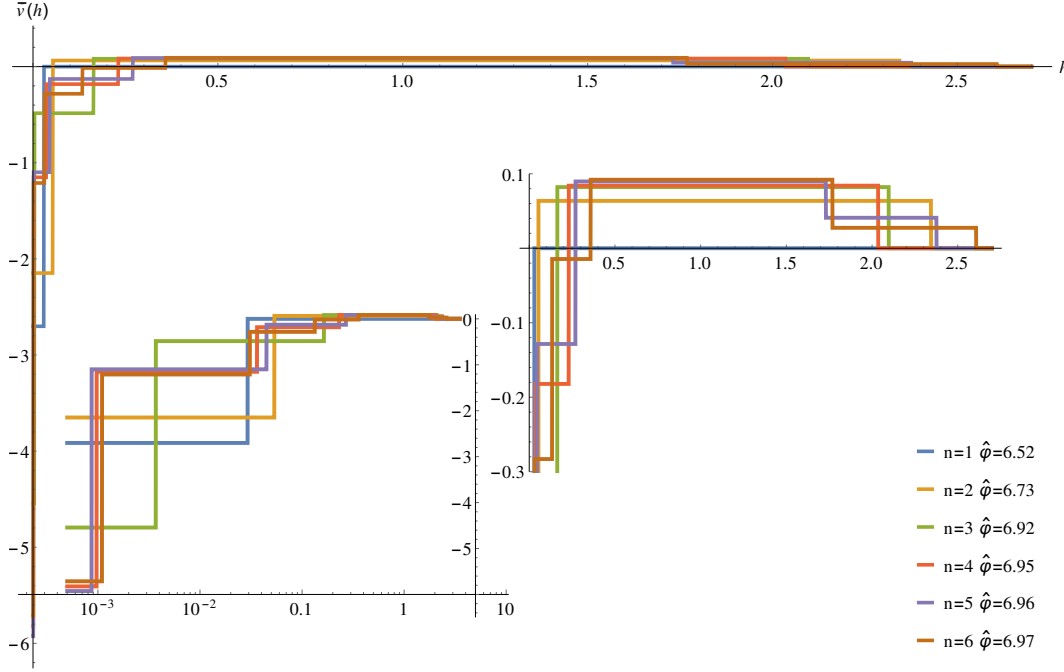

Figure 5: Plot of the different best potentials found by the gradient descent for $n = 1$ to 6. Inset on the left is zooming on the small distances logarithmically, while the right inset focuses on the tail of the potential.

# 5   Discussion

## 5.1   Summary of our results

In this paper, we have shown that one can endow hard-core spherical particles with a suitable extra soft interaction potential which allows to maximize the dynamical glass transition density $\varphi_{\rm d}$ of the packing. The optimal shape of the potential turns out to display a short-range attraction and weak repulsive tail. This amounts to bias the sampling measure of dense packings with a non-uniform distribution. By doing so we find that in a space of large dimension $d$ a set of disordered liquid-like hard-sphere configurations does exist up to a packing fraction of $\varphi_{\rm d} \approx 7d\,2^{-d}$ (see table 2 for the precise values). This set of configurations is equilibrated and it is possible to show that the configurational entropy density, which counts the number of distinct glass basins, is strictly positive at $\varphi_{\rm d}$. Interestingly, the first property implies that these configurations are thermodynamically stable against random errors in their algorithmic construction, while the second implies that the number of distinct glassy states, and of jammed configurations that can be constructed by compressing them, is exponentially large in the number of particles. Besides, we have explicitly checked that for all potentials in table 2, the excess entropy is $s_{\rm ex}(\varphi_{\rm d}) < 0$ from (5), which means that the second virial correction to the reduced pressure, dominant in large $d$ and given by $(\beta P/\rho) - 1 = -s_{\rm ex}(\varphi)$ [44–46] is positive and monotonously increasing with $\varphi$, thus forbidding an instability caused by phase separation.

Because the relaxation time of the equilibrium liquid is finite for $\varphi < \varphi_{\rm d}$, sampling equilibrium liquid configurations of $N$ particles requires a time of the order of $Nd$, the number of degrees of freedom in the system. This is because each degree of freedom must be updated a finite number of times (either by Monte Carlo or Molecular Dynamics) before equilibrium is reached. If the system of $N$ particles is confined in a box of linear size $L$ and volume $V = L^d$ with periodic boundary conditions, the resulting packing can be periodically extended to the infinite Euclidean space, for which it would be a periodic packing of period $L$ with $N \gg 1$ particles in the unit cell. For this procedure to be well defined, however, the linear size $L$ of the system must be at least $L = 2 + O(1/d)$, in such a way that a particle cannot collide with itself due to the periodic boundary conditions. Hence, the minimal number of particles must be of the order of

$$N \approx \frac{V\varphi_{\rm d}}{v_d} \approx 2^d \times \frac{d\,2^{-d}}{v_d} = 2^d \times \frac{d^2}{\Omega_d} \approx e^{\frac{d}{2}[\ln d - \ln(\pi e) + \ln 2]} \ . \tag{13}$$

where we used the relation $v_d = 2^{-d}\Omega_d/d$ for the volume of a particle of diameter 1, the Stirling approximation for the solid angle $\Omega_d \sim \exp[\frac{d}{2}\ln(2\pi e/d)]$, and we neglected all sub-exponential corrections. Note that this value of $N$ diverges with $d$, and because the finite size corrections vanish as $1/N$ in the liquid phase [49], in the large $d$ limit the resulting system is effectively in the thermodynamic limit despite the fact that its linear size[6] is $L = 2$. The results of our calculations can therefore be applied in this regime. The time needed to sample efficiently the liquid configurations, which scales proportionally to $dN$, is given by

$$\tau \approx dN \approx e^{\frac{d}{2}[\ln d - \ln(\pi e) + \ln 2]} \ , \tag{14}$$

---

[6]Note, however, that a $d$-dimensional cube of side $L = 2$ has $2^{d-1}$ diagonals of length $2\sqrt{d}$. Hence, its linear size along the diagonals is very large when $d \to \infty$.

where again we only kept the exponential dependency in $d$. Our main result can thus be summarized as follows:

*One can construct sphere packings of packing fraction $\varphi \approx 7d\,2^{-d}$ in a time scaling as in Eq. (14), by sampling equilibrium liquid configurations in a volume $V = (2 + O(1/d))^d$, using the pair potential given in table 2.*

We believe this statement to be exact in the sense of theoretical physics, even if of course we did not provide a mathematically rigorous proof. A reasonable hope that such a proof could be reached in the future relies on the many results that have been first obtained by the same heuristic statistical mechanics techniques we used and then rigorously confirmed (see for instance [71–73] for the static properties of mean-field spin-glasses, and [74, 75] for their dynamics). With respect to these works, which dealt with models with an intrisic mean-field character in their definition, an additional difficulty that would need to be overcome for a rigorous proof of the above statement is that the mean-field properties of interacting particle systems only arise when the large dimension limit is taken.

## 5.2    Comparison with related results

First, it should be noted that our result is very similar to the (rigorous) one of Moustrou [36], who proved that packings with density $\varphi \geqslant 0.89 \ln(\ln d) d\,2^{-d}$ can be constructed with $\exp(1.5\,d\ln d)$ binary operations, i.e. with the same form of scaling as in Eq. (14), although with a prefactor of 1.5 instead of 0.5 in the leading exponential term. While Moustrou's result has a better asymptotic scaling of the density for $d \to \infty$ ($\ln(\ln d)$ vs constant), our result is actually better as long as

$$d < e^{e^{7/0.89}} \approx 10^{1131} \; . \tag{15}$$

Because in practical applications one would never reach such high dimensions, we believe that our results remain of comparable interest to Moustrou's. Also, our method can produce an exponential number of distinct packings (the complexity, or configurational entropy, being extensive at the dynamic transition [3]), that would also be robust to thermal noise.

Second, it should be noted that the equilibrium packings produced by our method have finite pressures, hence the gaps between particles are positive. Such packings can thus be further compressed out of equilibrium, leading to an increase of the packing density. In the pure hard sphere case, this brings the density from $\varphi_{\mathrm{d}} \simeq 4.8067\,d\,2^{-d}$ to $\varphi_{\mathrm{j,d}} \approx 7.4\,d\,2^{-d}$. The potentials introduced in this paper have significantly increased the dynamic transition $\varphi_{\mathrm{d}}$, yet preliminary results seem to indicate that the gain on the final density after compression is quite weak. This could be rationalized through the following observation: the radial distribution function of our liquid packings, simply given here by $g(r) = e^{-v(r)}$ [67,76], exhibits oscillations reminiscent of the ones found for jammed configurations of hard spheres in $3 \leqslant d \leqslant 6$ [77,78]. The configurations we find here may well be quite close to jamming, although they belong to the equilibrated liquid with potential $v(r)$. We also emphasize that the jamming packing fraction $\varphi_{\mathrm{j,d}}$ obtained[7] by compression from $\varphi_{\mathrm{d}}$ does depend on the potential $\bar{v}(h)$, and more generally on the protocol used to compress the system [4], unlike the equilibrium SAT-UNSAT transition (or GCP).

---

[7]This packing fraction $\varphi_{\mathrm{j,d}}$ can be computed for any potential $\bar{v}(h)$ through the state-following procedure of [20].

Third, one should keep in mind that packings do exist up to at least $\varphi = 65963 \, d \, 2^{-d}$, according to the rigorous result of Venkatesh [35], and up to at least $\varphi \sim (\ln d) \, d \, 2^{-d}$, according to the non-rigorous results of [3]. However, it is possible that constructing these packings would take a much larger time than the one in Eq. (14).

## 5.3   Perspectives

There are several open problems to be explored. We now discuss some of them along with possible future directions of research.

1. Early work on the square-well interaction potential in infinite dimension has shown that our approach reproduces quite in detail the phase diagram of attractive colloids in finite dimension, previously found in the mode-coupling theory approach, numerical simulations and experiments [57–61]. The phase diagrams of these systems include non-trivial features such as reentrant behaviour, multiple glasses, and higher-order bifurcation singularity [66, 68]. The question that naturally arises is whether this correspondence is far more general and can be extended to more complex interaction potential as those studied in the present paper. The extension of the numerical approach to simulations of hard sphere systems up to $d = 13$ [49, 50] should be in principle straightforward and might be useful to clarify this important problem. This would also shed some light on the issue of the number of operations required to construct a dense packing, which appears still quite prohibitive as $d$ increases. For some related works in this direction on soft-core models, see [79, 80].

2. The actual design of an experimental soft-matter system interacting with a finite-dimensional analog of our optimized potential (displayed in table 2 and figure 5) might be a challenge to experimental physicists. While the simple square-well potential ($n = 1$) can be seen as a limiting case of the entropy driven (Asakura-Oosawa) depletion force in a strongly asymmetric binary mixture, it is not obvious that something similar can be achieved for a generic $n$-step interaction potential. Nevertheless, focusing on the relatively simpler $n = 2$ or $n = 3$ potentials displayed in table 2 would be already much interesting. In fact, some cases of this type have been already addressed quite extensively at moderate packing fractions, and they are known to exhibit intriguing features such as liquid-like anomalies and multiple liquid phases [53–55], and instabilities producing disordered or periodic microphases [63–65]. Our work shows that complex glassy features should appear in these systems at higher packing fractions, provided that the extension to finite dimension holds true.

3. From our data one surmises that a two-body interaction potential with spherical symmetry and with $n$ steps (wells or shoulders) should have $n + 1$ distinct glass phases, each of which is associated to a privileged packing arrangements over a suitable length scale dictated by the range of wells/shoulders and depths/heights entering the potential (notice that the $+1$ comes from the short range hard core repulsion). Therefore, we naturally expect that the interplay of multiple glass phases gives rise to several glass-glass transitions and higher-order critical points, as they occur in simpler schematic models [81, 82]. In fact, in our search of the optimum potential we find many such features (see Figure 4), although we did not study them in detail. From a dynamical point of view we believe that the high-density liquid phase should exhibit very special

physical properties and novel relaxation patterns, in particular, near the tip of the reentrancy region, which is surrounded by several competing glass phases. For this reason one can arguably expect that the time correlation decay features a multi-step relaxation mechanism (as far as aging dynamics is concerned, that would possibly correspond to a multi-step effective temperature scenario after a subcritical quench).

While most of the above perspectives concerns applications in soft matter, from the point of view of packing the challenge of proving rigorously our results remain, of course, an extremely important one.

# Acknowledgements

We thank Elliot Menkah for the parallelization of the code, and Patrick Charbonneau, Will Perkins and Antonello Scardicchio for discussions. The numerical calculations have been performed by using the clusters available at LPT-ENS and ICTP.

**Funding information** – This project has received funding from the European Research Council (ERC) under the European Union's Horizon 2020 research and innovation programme (grant agreement n. 723955 - GlassUniversality).

# A  The algorithm

The gradient descent algorithm is implemented as follows:

1. Start from some point $M$ in parameter space for a fixed discretization $n$ of the potential. In practice the initial conditions are chosen by taking approximately 50-70 (for $n \leqslant 5$) highest values of the packing fraction out of a loose sampling of the $2n$-dimensional parameter space. Due to the constant signs of the values of the amplitudes $U_i$ throughout a given descent, one must also check that all relevant sign combinations are present in this ensemble of initial conditions. As already mentioned, this is done in order to sample all possible local maxima of $\widehat{\varphi}_{\mathrm{d}}$, if any.

2. Compute the function $\mathcal{F}(\Delta|M)$ for a fine enough discretization of $\Delta$ and store the positions of its local maxima. The $y$ integrations appearing in computations involving $\mathcal{F}$ are performed on a suitable interval where $\mathcal{F}$ is not vanishingly small and using Simpson's interpolation.

3. This function has a certain number of local maxima, in particular there are $m-1$ maxima which are closer to the global one than some predefined small value. If $m = 1$, start the update with the standard gradient descent. If $m > 1$, one must use the modified gradient descent within the GG manifold, noted $\mathscr{L}$, see the next appendix.

4. Compute the update, i.e. the term proportional to $\varepsilon$ by calculating the gradients in the unique global maximum (for the standard gradient descent) or the modified direction along $\mathscr{L}$ (for the improved gradient descent).

5. To optimize the update, a linear search is performed: the values of $\widehat{\varphi}$ with the updated parameters are computed for several decades of $\varepsilon$ (i.e. for several points in logarithmic scale on the line given by the chosen direction) and the highest value of the packing fraction is kept as the updated value in parameter space, which constitutes the new estimate of the parameters $\boldsymbol{M}$. The algorithm ends if no value of the packing fractions computed within the linear search is higher than the one given by the parameters before update. This linear search is parallelized to reduce run time.

# B The gradient descent within the transition manifolds between glass phases

Here we provide more details on the direction the gradient descent algorithm has to choose when $m \geqslant 2$ maxima of $\mathcal{F}(\Delta|\boldsymbol{M})$ become equal. We recall that the point $\boldsymbol{M} = (U_0, \sigma_0, \ldots, U_{n-1}, \sigma_{n-1})$ represents the potential parameters.

In the following we assume that the algorithm arrives close to a region where $m \geqslant 2$ maxima of $\mathcal{F}(\Delta|\boldsymbol{M})$ become equal. We note the value of $\mathcal{F}$ in each of the local maxima $\Delta_i(\boldsymbol{M})$, $f_i(\boldsymbol{M}) := \mathcal{F}(\Delta_i(\boldsymbol{M})|\boldsymbol{M})$ and define the manifold of the transition

$$\mathscr{L} = \{\boldsymbol{M} \,|\, f_1(\boldsymbol{M}) = f_2(\boldsymbol{M}) = \ldots = f_m(\boldsymbol{M})\} \tag{16}$$

Now we would like to perform a gradient descent inside this manifold. Let us take $(\boldsymbol{M}, \boldsymbol{M}') \in \mathscr{L}^2$ with $\boldsymbol{M}' = \boldsymbol{M} + \delta\boldsymbol{M}$. We have for each maximum of $\mathcal{F}$:

$$f_i(\boldsymbol{M}') = f_i(\boldsymbol{M}) + \delta\boldsymbol{M} \cdot \boldsymbol{\nabla} f_i(\boldsymbol{M}) + O(\|\delta\boldsymbol{M}\|^2) \tag{17}$$

Note that, a priori, these functions have no singularities unlike the gradient of $\widetilde{\mathcal{F}}(\boldsymbol{M})$. By substracting any two equations we get that $\delta\boldsymbol{M}$ is locally (i.e. at first order) orthogonal to any difference of gradients $\boldsymbol{\nabla} f_i(\boldsymbol{M}) - \boldsymbol{\nabla} f_j(\boldsymbol{M})$. The vector space spanned by these is generated by $\{\boldsymbol{e}_1, \ldots, \boldsymbol{e}_{m-1}\}$ where $\boldsymbol{e}_i = \boldsymbol{\nabla} f_i(\boldsymbol{M}) - \boldsymbol{\nabla} f_{i+1}(\boldsymbol{M})$ if we assume the family of vectors $\{\boldsymbol{e}_1, \ldots, \boldsymbol{e}_{m-1}\}$ to be linearly independent. Under this condition, $\mathscr{O}(\boldsymbol{M}) = \text{Span}(\boldsymbol{e}_1, \ldots, \boldsymbol{e}_{m-1})$ is thus the local orthogonal subspace to the manifold in which we wish to perform the gradient descent. We can uniquely expand the gradients as

$$\boldsymbol{\nabla} f_k(\boldsymbol{M}) = \boldsymbol{a}_k + \sum_{i=1}^{m-1} \lambda_i^k \boldsymbol{e}_i \tag{18}$$

where $\boldsymbol{a}_k \in \mathscr{O}(\boldsymbol{M})^\perp$ is tangent to the manifold $\mathscr{L}$. Since $\delta\boldsymbol{M} \cdot \boldsymbol{\nabla} f_k(\boldsymbol{M}) = \delta\boldsymbol{M} \cdot \boldsymbol{a}_k$ the minimization of $f_k(\boldsymbol{M})$ inside the manifold, from (17), locally follows the direction of $-\boldsymbol{a}_k$. Since the newly reached point $\boldsymbol{M}'$ is also in $\mathscr{L}$, all other maxima will have the same value, it is thus also a direction that decreases $\widetilde{\mathcal{F}}$.

Next, we notice that $\forall(i,j) \in [\![1, m-1]\!]^2$, $\forall\delta\boldsymbol{M} \in \mathscr{O}(\boldsymbol{M})^\perp$, we have $\boldsymbol{a}_i$ and $\boldsymbol{a}_j$ also in $\mathscr{O}(\boldsymbol{M})^\perp$, and $\delta\boldsymbol{M} \cdot \boldsymbol{a}_i = \delta\boldsymbol{M} \cdot \boldsymbol{a}_j$, hence actually[8] $\boldsymbol{a}_i = \boldsymbol{a}_j$. This means we can take any $-\boldsymbol{a}_k$ to compute the correct direction. This direction is easy to compute since $\boldsymbol{a}_k = \boldsymbol{\nabla} f_k(\boldsymbol{M}) -$

---

[8]This has also been numerically checked.

$\sum_{i=1}^{m-1} \lambda_i^k \boldsymbol{e}_i$. The coefficients are obtained solving a linear equation [83], projecting (18) onto each $\boldsymbol{e}_j$. One has

$$\lambda_i^k = \sum_{j=1}^{m-1} Q^{-1}{}_{ij} \boldsymbol{\nabla} f_k(\boldsymbol{M}) \cdot \boldsymbol{e}_j \tag{19}$$

where $Q$ is the symmetric matrix of the overlaps[9] $Q_{ij} = \boldsymbol{e}_i \cdot \boldsymbol{e}_j$.

---

[9] If the dimension of the vectors is 1, this matrix is singular. If not, which is the case here since the gradients live in a $2n$-dimensional space, there is no a priori reason for a singularity.

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
