# Peer review of "Generating dense packings of hard spheres by soft interaction design"

_SciPost Physics_

## Round 1 · Referee Report · Anonymous · 2018-3-14

Strengths

1-The basic idea of the paper is original and useful in practice.
2-The results are interesting and sound.
3-The presentation is quite pedagogical and easy to follow.

Weaknesses

1-The paper could be streamlined and made shorter.

Report

As said above, I deem the ideas proposed in the paper to be original and the results to be interesting and sound, certainly fit for publication on this platform. However, I have a concern. In the introduction, the authors write that "dressing" the hard-core repulsion with a potential tail is not supposed to influence the SAT-UNSAT transition, which is a reasonable statement. However, in paragraph 5.2 they mention that, once the optimized potential is added to the interaction, the jamming density of the configurations produced at its associated dynamical transition, $\varphi=\varphi_d$, increases (if only a little), when the optimized potential is added to the interaction. This means that all the out-of-equilibrium packings with a jamming density between the bare hard-sphere one ($\hat{\varphi_j,d} \simeq 7.4$) and the new one obtained from the optimized potential, are cut out of the J-line and supposedly unreachable by adiabatic compression from the liquid, despite the potential tail being supposedly (and reasonably) irrelevant on the J-line.
I'd like the authors the elaborate on this (both in their answer to this report and in the paper), before publication can be recommended.

Requested changes

I'd like the authors to address the concern described in the report, before I can recommend publication of the manuscript.

---

## Round 1 · Referee Report · Anonymous · 2018-3-26

Strengths

1) The paper is very interesting, sound and well written;
2) I also appreciate the appendix section, where more details on the algorithmic protocol can be found, and the final comparison with very recent rigorous results on the same subject;
3) This work can give important insights on finite-dimensional systems as well, as the authors themselves highlight in the perspectives.

Weaknesses

No weak point to mention.

Report

The authors study the challenging theoretical problem of sphere packings, which dates back to Kepler, in a statistical mechanics approach.
With respect to a purely hard-sphere system in high dimensions, they bias the uniform Gibbs measure by adding an attractive potential, which, in the simplest case, is also known as square-shoulder potential in the colloids’ literature.
After a very useful recap of the main results known thus far concerning the critical packing fraction bounds and the relations in the high-dimensional limit between the potential method and the dynamical density, they address the problem numerically, via both a “brute force” search and a modified gradient descent method.
In the presence of a short-range attractive potential, a reentrance transition between the liquid and the glass phases appears, as two of the authors already noticed in two previous related works.
The main aim of this paper is to achieve a better understanding of this reentrance transition and to extend it, by solving the inverse problem of maximizing the dynamical density over the parameter space of a trial family of potentials. The numerical analysis is performed using piecewise constant potentials with a given number of steps, from n=1 to n=6.
They describe in detail both the single-step case and its generalization to a higher number of steps.
However, going beyond the n=5 case does not provide any particular improvement since in any case the packing fraction saturates close to the value 7d2^{-d}.

As I wrote in the first section above, I find the paper very interesting and extremely clear, with numerous potential applications in finite-dimensional condensed matter physics.
I thus recommend the paper for immediate publication.

Requested changes

No requested changes.

I have only a personal suggestion that I leave to the authors' choice.
Since they mention possible connections with experiments and colloidal material setups in the conclusions, Sec. 5.3, I would suggest to include some references while mentioning the “square-shoulder potential” already in Sec. 3.3 (first paragraphs) to better remark why this potential can be of interest in related fields.

---

## Round 1 · Referee Report · Anonymous · 2018-3-29

Strengths

1. The introduction is very pedagogical
2. The work has high originality

Weaknesses

1. Outcome is not impressive

Report

The authors study the random packing problem in the infinite dimensional limit. Practically, construction of hard sphere packings in higher dimensions is very difficult task because it is hampered by the dynamic transition. The authors attempt to bypass this difficulty by tuning biased potentials so that the dynamic transition is pushed toward higher density. This inverse problem is numerically performed thanks to the exact solution for the dynamic transition. This work shows high originality and provides some interesting physical insights such as connection between the number of different glass phases and relevant lengthscales in a given pair potential. The work would be useful for a wide range of fields, e.g., statistical physics, information science, and soft matter.

I recommend the draft for publication. However I would like to ask the authors the following questions and comments.

1) Is there a gain eventually?
I think the original goal is obtaining high density hard sphere packings (unbiased system) more effectively. The biased potential system have higher dynamic transition density than the unbiased system, which enables us to access higher density equilibrium configurations for the “biased” system. Starting from such a configuration, once the biased potential is turned off, the configuration is not in equilibrium anymore for the “unbiased” system. Therefore, subsequent further compression would not produce higher density jammed packings. In other words, it is not guaranteed that the biased potential system provides well-annealed configurations for the pure hard sphere system. I think this reasoning explains the fact that the authors did not find much denser jammed packing (inherent structure of the unbiased system) after the compression of the biased system.

2) Finite excess entropy
The authors mention that the obtained packings are thermodynamically stable because of finite excess entropy. Conventionally speaking, thermodynamic stability is related to the convexity of the entropy or its derivative, thus the absolute value is meaningless. Therefore the authors should add more words for the role of finite excess entropy in the context.

3) Another functional form for the biasing potential
This is just a suggestion. The authors found that the shape of the best-packing potentials consists of a sticky attraction and a positive tail. This kind of potential form can be more effectively explored by the Jagla potential which is a model of water. This model has 4 parameters (n=2), thus one could easily study by full sampling.

Requested changes

1) Typo
There is no overline on the label of Y-axis.

---

## Round 2 · Referee Report · Anonymous · 2018-5-7

Report

I deem the authors' response to my concerns to be satisfactory enough, and I therefore recommend publication of the manuscript in its present form.

---

## Round 2 · Referee Report · Anonymous · 2018-5-8

Report

The authors have satisfactorily addressed my (optional) concerns in the first
report. This is a solid and useful paper, it will be surely appreciated by researchers in the field.

---

## Round 2 · Referee Report · Anonymous · 2018-5-11

Report

The authors answered all my questions convincingly and added a more explanation related to my question to the text. Thus I recommend the current version for publication.

---

## Round 2 · Author Response

Dear Editor,

we warmly thank the referees for their positive appreciation of our work and for their insightful comments that have helped us to improve our manuscript in this revised version. We provide below answers to the referee's questions, as well as a summary of the changes performed.

Best wishes,
The authors
* * *
REPORT 1

-The paper could be streamlined and made shorter.

We unfortunately don’t see exactly how to make the paper shorter without loosing relevant information.

QUESTION:

As said above, I deem the ideas proposed in the paper to be original and the results to be interesting and sound, certainly fit for publication on this platform. However, I have a concern. In the introduction, the authors write that "dressing" the hard-core repulsion with a potential tail is not supposed to influence the SAT-UNSAT transition, which is a reasonable statement. However, in paragraph 5.2 they mention that, once the optimized potential is added to the interaction, the jamming density of the configurations produced at its associated dynamical transition, φ=φd, increases (if only a little), when the optimized potential is added to the interaction. This means that all the out-of-equilibrium packings with a jamming density between the bare hard-sphere one (^φj,d≃7.4) and the new one obtained from the optimized potential, are cut out of the J-line and supposedly unreachable by adiabatic compression from the liquid, despite the potential tail being supposedly (and reasonably) irrelevant on the J-line.
I'd like the authors the elaborate on this (both in their answer to this report and in the paper), before publication can be recommended.

ANSWER:

Let us begin by stressing that when we write, in the introduction, that the SAT-UNSAT transition is unaffected by the potential, we mean the equilibrium phase transition where no (disordered) packing exist, which corresponds to the Glass Close Packing φ_GCP. It can be shown analytically that the location of this transition only depends on the hard core, and not on the finite part of the potential.

On the contrary, the lower limit φ_{j,d} of packings that can be reached by out-of-equilibrium compression from an equilibrated liquid with a given potential v(h), i.e. the J-line associated to a given potential, will certainly depend on v(h). This is to be expected because in general the J-line is protocol dependent.

We added a remark on this in section 5.2.
We do not want to elaborate too much on this point because it would require introducing the equations that determine both φ_GCP and φ_{j,d}, and explain what happens during the state following from φ_d to φ_{j,d}, which also requires the introduction of fullRSB ideas.
Our paper is instead focused on the much simpler problem of maximising φ_d, which requires much less mathematics to be discussed.
* * *
REPORT 2

I have only a personal suggestion that I leave to the authors' choice.
Since they mention possible connections with experiments and colloidal material setups in the conclusions, Sec. 5.3, I would suggest to include some references while mentioning the “square-shoulder potential” already in Sec. 3.3 (first paragraphs) to better remark why this potential can be of interest in related fields.

ANSWER:
We added more references in the beginning of section 3.1, as suggested by the referee.
* * *
REPORT 3

Weaknesses
1. Outcome is not impressive

We agree that unfortunately the increase in lower bound is modest. Still in our opinion the physical mechanism is general and could be interesting both in this and in other related fields.

QUESTION 1) Is there a gain eventually?
I think the original goal is obtaining high density hard sphere packings (unbiased system) more effectively. The biased potential system have higher dynamic transition density than the unbiased system, which enables us to access higher density equilibrium configurations for the “biased” system. Starting from such a configuration, once the biased potential is turned off, the configuration is not in equilibrium anymore for the “unbiased” system. Therefore, subsequent further compression would not produce higher density jammed packings. In other words, it is not guaranteed that the biased potential system provides well-annealed configurations for the pure hard sphere system. I think this reasoning explains the fact that the authors did not find much denser jammed packing (inherent structure of the unbiased system) after the compression of the biased system.

ANSWER:
The goal is to obtain well-defined sphere packings where the spheres do not overlap, which is needed for the mathematical definition of a sphere packing, as dense as possible. The packings produced via the biased system are nevertheless valid packings in these sense (of higher packing fraction), since the spheres do not overlap.

This said, it is true that the equilibrium configurations of the biased systems reachable below the new dynamical transition are certainly not equilibrium configurations of the pure hard-sphere system. We agree that from the point of view of the density of the final, out-of-equilibrium configurations reachable by a further compression of the system the gain is not impressive, however we believe that the intermediate result on the modification of the dynamic transition by the addition of the biasing potential could be interesting by itself.

Moreover, obtaining analytical results for the out-of-equilibrium compression requires quite sophisticated methods (fullRSB state following technique), while the calculation of the dynamical transition is straightforward. In this sense there is much more hope to make the results for φ_d rigorous, with respect to the one for φ_{j,d} (see also the answer to referee 1 above).

QUESTION 2) Finite excess entropy
The authors mention that the obtained packings are thermodynamically stable because of finite excess entropy. Conventionally speaking, thermodynamic stability is related to the convexity of the entropy or its derivative, thus the absolute value is meaningless. Therefore the authors should add more words for the role of finite excess entropy in the context.

ANSWER:
We meant that the packings are thermodynamically stable in the sense that they are equilibrated configurations, so they are by definition robust to thermal noise.
The fact that the entropy is finite means that each packing has a lot of other packings in its vicinity, which means that a little perturbation does not disrupt the property of being a valid hard sphere configuration.
Besides, we also checked that the first virial correction to the equation of state is positive (negativity would imply instability towards phase separation, which may happen for a liquid potential with an attractive component).
We added these clarifications in the paper, at the beginning of section 5.1.

QUESTION 3) Another functional form for the biasing potential
This is just a suggestion. The authors found that the shape of the best-packing potentials consists of a sticky attraction and a positive tail. This kind of potential form can be more effectively explored by the Jagla potential which is a model of water. This model has 4 parameters (n=2), thus one could easily study by full sampling.

ANSWER:
This is actually something we have tried. In this paper we discretized the potential into constant steps, which notably renders the computation of the function q(\Delta,y) in equation (7) easier numerically. But one notes that instead of constant steps, one may as well take linear ramps or parabolic piecewise discretization, both would yield a simple expression of q in terms of error functions and exponentials.
Moreover, these would converge quicker onto a limit shape than the stepwise discretization.
However, already for a single linear ramp v(h)=a*h+b for h>\sigma, v(h)=0 for h >sigma (3 parameters), we were not able to sample the full phase space phi_d(a,b,\sigma). Although some parts of it were fine, generally as soon as b gets large (>10 or 20) we had many numerical errors.
The typical value of b here from a fit of the constant steps potentials obtained would be ~150. One can understand the numerical errors coming from this large value inside exponentials (also the small range of \sigma gives issues). We couldn't solve this even by using asymptotic functions (such as erfcx) or expansions. We thus decided to stick with the better behaved stepwise discretization.

Note also that the shift of φ_d when moving from n=4 to n=6 is very modest, which implies that the result is likely to be converged to the asymptotic limit n=oo, which must be independent of the discretization (we just expect that having an infinite number of steps will smooth the steps while conserving the shape of it, and marginally increasing the associated dynamical packing fraction).
* * *

---

## Round 2 · List of Changes

Please see the comments section.
The few changes are in sections 3.1 and 5.
Several citations have been added [55,56,62-65,77,78,81-83].

---

## Editorial Decision

ontology_/_topics